# Effects of Plant-Emitted Monoterpenes on Anxiety Symptoms: A Propensity-Matched Observational Cohort Study

**DOI:** 10.3390/ijerph20042773

**Published:** 2023-02-04

**Authors:** Davide Donelli, Francesco Meneguzzo, Michele Antonelli, Diego Ardissino, Giampaolo Niccoli, Giorgio Gronchi, Rita Baraldi, Luisa Neri, Federica Zabini

**Affiliations:** 1Department of Medicine and Surgery, University of Parma, I-43121 Parma, Italy; 2Division of Cardiology, Azienda Ospedaliero-Universitaria di Parma, I-43126 Parma, Italy; 3Institute of Bioeconomy, National Research Council, 10 Via Madonna del Piano, I-50019 Sesto Fiorentino, Italy; 4Central Scientific Committee, Italian Alpine Club, 19 Via E. Petrella, I-20124 Milano, Italy; 5Department of Public Health, AUSL-IRCCS of Reggio Emilia, I-42122 Reggio Emilia, Italy; 6Section of Psychology, Department of Neuroscience, Psychology, Drug Research and Child’s Health (NEUROFARBA), University of Florence, 12 Via di San Salvi, I-50135 Firenze, Italy

**Keywords:** anxiety, biogenic volatile organic compounds, forest therapy, monoterpenes, α-pinene, propensity matching

## Abstract

Immersive experiences in green areas, particularly in forests, have long been known to produce beneficial effects for human health. However, the exact determinants and mechanisms leading to healthy outcomes remain to be elucidated. The purpose of this observational cohort study was to investigate whether inhaling plant-emitted biogenic volatile compounds, namely monoterpenes (MTs), can produce specific effects on anxiety symptoms. Data from 505 subjects participating in 39 structured forest therapy sessions at different Italian sites were collected. The air concentration of monoterpenes was measured at each site. STAI state questionnaires were administered before and after the sessions as a measure of anxiety. A propensity score matching analysis was then performed, considering an above-average exposure to inhalable air MTs as the treatment. The estimated effect was −1.28 STAI-S points (95% C.I. −2.51 to −0.06, *p* = 0.04), indicating that the average effect of exposure to high MT air concentrations during forest therapy sessions was to decrease anxiety symptoms.

## 1. Introduction

The preventive and healing effects of green environments on psychophysical well-being have been documented extensively in scientific literature and numerous studies suggest that forest exposure is associated with a wide range of benefits, covering both psychological [1,2,3,4,5], and physiological aspects [6,7,8,9]. In particular, a few experimental studies have demonstrated the effects of forest immersion in stress mitigation and the induction of physiological relaxation, especially with regard to anxiety levels [10,11,12,13,14], a topic analyzed in recent literature reviews [5,15,16].

The beneficial effects of visiting a forest derive from the combined stimulation of all senses by specific features of the natural environment [7] whose impact and quantitative assessment have not yet been fully elucidated. Biogenic volatile organic compounds (BVOCs) emitted by plants and soil in the forest atmosphere, and in particular monoterpenes (MTs), as key constituents of BVOCs, have often been suggested as one of the determinants of the interaction between forest ecosystems and human health [17,18,19,20], especially those related to long-lasting effects on the immune system after forest exposure [18,21,22,23,24,25].

All plant organs, from flowers to roots, can generate and release MTs [26], with leaves generally responsible for the highest emission rates [27]. MT emission rates and chemical profiles vary widely among different species of forest trees [28,29] and they depend on physiological (plant age and developmental stage) and physicochemical factors, i.e., related to leaf structure (presence or absence of storage structures and stomatal openness) [30]. The individual perception of benefits induced by plant-emitted MTs has stimulated research efforts to expedite the modeling and prediction of forest emissions [31,32], along with preliminary studies aimed at predicting nose-level concentrations of BVOCs [33]. Important efforts have also been made to design forest therapy programs and trails based on potential emissions of BVOCs released by dominant plant species [34].

Recently, several studies aiming at assessing and quantifying BVOCs in forest sites have been published [20,31,32,35,36], owing to the need to gain further insights into forest characteristics potentially related to specific effects on health outcomes [34,37,38,39,40,41,42].

Overall, the evidence about MT activity on human health mostly derives from in vitro, animal, or indoor/laboratory experiments, usually with a small number of study participants [43], and very few studies take into consideration MT concentrations along with the measurement of health outcomes [23,44]. Only recently, a few studies investigated the most important psychophysiological short-term effects of individual exposure to plant-emitted MTs. However, such studies were limited to specific and/or numerically scarce target groups, for example, maladjusted soldiers [45], or to specific plant species, such as *Cinnamomum burmannii*, growing in partially controlled environments [46]. In general, even though the results of these studies are encouraging, quantitative experiments in natural environments are still lacking.

There is also a dearth of studies investigating any specific health effects of terpenes inhaled during forest exposure and their mechanism of action from a pharmacological point of view [47]. Only two studies measured the serum concentration of monoterpenes after forest exposure, reporting a sixfold increase of α-pinene levels after walking in a conifer forest for one hour [48] and a higher absorption of α-pinene after walking in an evergreen broadleaf forest for two hours, but only in participants with lower baseline blood concentrations of monoterpenes [35]. Although relevant, the impact of the latter two studies was affected by small sample sizes. Moreover, some preclinical evidence exists from animal studies of the possible anxiolytic activity of α-pinene [7,49,50,51].

Anxiety is a highly prevalent nonpsychotic mental disorder, and one of the most increasing psychiatric disorders of the last decades. Anxiety disorders have a worldwide prevalence of up to 15% in the general population and are more common in women [52,53]. The global burden of hypertension is extensively impacted by anxiety, both as a risk factor and as an exacerbating factor [54]. Anxiety is also a known risk factor for major adverse outcomes in heart diseases, especially in heart failure, chronic and acute coronary syndromes, and cardiomyopathies [55,56,57,58]. Finding new sustainable strategies to decrease the burden of anxiety symptoms is an important goal for researchers and would have a significant impact on health policies.

The aim of this study was to assess whether exposure to inhalable MTs during 3-h long forest therapy sessions can produce specific effects on anxiety levels. This study follows a previous pilot study, where the total concentration of BVOCs was preliminarily suggested as a contributing factor associated with short-term improvements in mood states, in particular with anxiety reduction in subjects involved in forest therapy sessions organized in remote green areas [14]. To the authors’ best knowledge, this is the first attempt, grounded on a large number of participants, to clarify the extent of the effect on anxiety levels of MTs inhaled in a natural setting.

## 2. Materials and Methods

### 2.1. Design and Participants

This research was originally designed as an observational cohort study. Experimental sessions were conducted between September 2021 and October 2022 at 39 sites, including 27 remote mountain forest sites, 3 hill forest sites, 1 remote coastal pinewood forest site, 7 urban parks in Italy, and 1 remote mountain forest site in Carinthia, Austria, located at altitudes between 5 m a.s.l. and 1800 m a.s.l. Figure 1 shows the location of these sites. During each session, one or more groups of participants, each made up of about 15–20 people, were guided along identified paths by trained operators, for a total duration of 3.0 ± 0.5 h. Data were collected immediately before and after each forest therapy session, as in pre–post studies without a follow-up period.

Only adults being able to autonomously fill in a questionnaire written in Italian and were willing to participate in forest therapy sessions were included in this study. Exclusion criteria were minor age (<18 years), any acute illnesses requiring medications other than those usually taken on a daily basis for chronic diseases, and any major physical or mental health problems incompatible with outdoor walking for a few hours. Participants were allowed to attend only one session, to prevent any possible carry-over effect.

Only participants who gave their informed consent were included in the data collection. All participants were treated in accordance with the ethical guidelines for research provided by the Declaration of Helsinki and its revisions, and the study was approved by the local ethics committee (CNR Ethics Committee n. 0069654/2021).

### 2.2. Outcome Measures

All participants autonomously reached each forest therapy site, and upon arrival, they were required to fill in the informed consent form. Then, every participant was presented with an anonymous questionnaire (with a unique ID code), which was aimed at collecting information about gender, age, place of residence, job, height, weight, chronic diseases, allergies, medications used, smoking habit, sports activities, and any previous experience in meditation practices. Moreover, the participants were asked to rate their subjective satisfaction in their relationships with friends and relatives with a 10-point Likert scale. Then, they were asked to fill in two psychometric questionnaires, the State-Trait Anxiety Inventory (STAI) and the Profile of Mood States (POMS) questionnaires.

The STAI questionnaire is widely used to measure state and trait nondisorder-specific anxiety, in both healthy and clinical populations [59]. It consists of a 40-item self-report measure of anxiety, using a 4-point Likert-type scale for each item. It has two scales: state anxiety, i.e., how one feels at a given moment (STAI-S), and trait anxiety, i.e., how one generally feels (STAI-T), both consisting of 20 items. Total scores were calculated by summing up the items in each respective form of anxiety (ranging from 20 to 80), where higher scores reflect greater levels of anxiety. The POMS questionnaire consists of 40 items, each measured with a 5-point Likert scale, related to domains such as tension, depression, anger, vigor, fatigue, confusion, and self-esteem [60]. Although both the STAI-S and the POMS questionnaires are widely used to explore different domains, the STAI-S was chosen as the outcome measure of state anxiety, i.e., the outcome measure of this study. Unlike the POMS, the STAI-S questionnaire is often used with clinical and subclinical study populations, and it can detect subtle changes in anxiety levels, even when there are large differences in baseline scores, which occur in groups differing in their vulnerability to anxiety, as is the case with highly heterogeneous participants to forest therapy sessions [61]. 

The STAI-S questionnaire was administered immediately before and after each forest therapy session in order to intercept any short-term effects (changes from baseline) elicited during the sessions. Change-from-baseline STAI-S scores reported by the study participants were considered the main outcome measure.

The STAI-T questionnaire was administered only at the baseline to better characterize the participants on the basis of their trait anxiety, which is a stable long-term tendency to anxiety. For a more comprehensive evaluation, the POMS questionnaires were also collected at baseline in order to characterize the participants in terms of other baseline mood states, i.e., nonanxiety-related POMS domains.

### 2.3. Exposure

The study participants were involved in forest therapy sessions aimed at maximizing their exposure to air MTs for an amount of time considered sufficient to be absorbed systemically and possibly exert a specific pharmacodynamic effect.

A professional psychologist or psychotherapist took part in each of the forest therapy sessions. The conduction protocol consisted of simple instructions to the participants to focus one’s attention on the external environment. The protocol was chosen to give reproducible instructions for all the psychologists or psychotherapists involved, in order to minimize any interferences related to the therapists’ personal styles and to achieve the most homogeneous experimental conditions possible.

The overall duration of each session was approximately 3-h, including short, slow walks, interspersed with five stops, and a final walk, before filling out the STAI-S questionnaires after the forest therapy experience. The physical intensity of the sessions was kept low to avoid possible biases due to adrenergic hyperactivation. Each session was performed in the morning, starting around 10:00 a.m. and ending around 1:00 p.m., local time (corresponding for all sessions to GMT+2). Forest therapy sessions were performed only if the weather was at least fair, without rain, and with comfortable temperatures within the range of 12 to 26 °C, collected by hand-held thermometers at the BVOCs sampling sites. The standardization of the method of conducting all forest therapy sessions, along with the same timing, was aimed at reducing possible biases.

#### Measurement of Biogenic Volatile Organic Compounds

The total and individual concentrations of different BVOCs in each forest site were measured through 1-h air samplings performed at 9.00 a.m., 11.30 a.m. and 2.00 p.m. Air was passed through the adsorption traps by using portable pumps (Pocket389Pump, SKC Inc., Washington County, PA, USA), that provided a constant flow of air of 200 mL/min, yielding sampling volumes of 12 l per hour.

Traps were made of inert metal tubes (8 cm × 0.3 cm i.d.) filled with Tenax TA and Carbograph 1TD (350 mg; 35/60 and 40/60 mesh, respectively), provided by Markes International, Ltd. (Llantrisant, UK), and they were stored at −20 °C until the analysis. BVOCs, released from traps using a thermal-desorption unity series 2, (Markes International, Sacramento, CA, USA), were injected into a 60 m capillary column (HP-1, 0.25 mm I.D.) internally coated with a 0.25 μm film of polymethylsiloxane, provided by J&W Scientific USA, Agilent Technologies (Palo Alto, CA, USA). BVOC separation was performed on a 7890A gas chromatograph and eluted compounds were detected with a 5975C mass spectrometer (GC–MS, Agilent Technologies, Wilmington, DE, USA).

For BVOC identification, retention times and mass spectra were collected and compared with the NIST 11 library, using the Agilent MassHunter Qualitative Analysis software. After identification, the compounds were quantified with an external standard calibration approach, using a gas cylinder with predefined concentrations of different BVOCs (isoprene and α-pinene; manufacturer: Apel-Riemer Environmental Inc., Broomfield, CO, USA). The precision of BVOC analysis varied among different chemical species and was generally characterized by a limited percentage range (2–5%). Blank emissions and artifact formation, which can affect the sensitivity and overall performance of this method, were determined with clean tubes. The detection limit of BVOCs was commonly achieved at the 0.1 ng level. Relative response factors were determined for all components, and for a minority of them, interpolation techniques were used. The concentration of each compound sampled in the experimental sites was calculated in µgm^−3^ as the mean and standard error of the three replicates [62,63,64].

Calibration curves were obtained with standard compounds (Sigma–Aldrich Chemical Co. St Louis, MO, USA). The concentration of each compound was calculated in µgm^−3^ as the mean and standard error of the three replicates. Figure 2 shows the calibration curve for α-pinene as one of the most representative MTs.

Among BVOCs, the air concentrations of MTs α-pinene, camphene, o-cymene, and sabinene were examined. For the purposes of the study, the concentration of MTs was measured at 11.30 a.m. at each site, because that timing was the most representative of the sessions. The total air concentration of MTs was selected as the principal exposure factor.

### 2.4. Predictors and Confounders

Possible predictors and confounders for a specific effect of MT exposure on anxiety symptoms were chosen among the different demographic, physiologic, psychologic, and sociologic domains in order to take advantage of the large sample size recruited for this study, to choose a good comparison between groups and, therefore, to intercept any possible specific effect of the exposure to inhaled MTs. The peculiarity of the propensity-score matching is indeed the chance to group individuals exposed and unexposed to a specific intervention by forming “couples” of subjects who were as similar as possible for all the covariates included in the propensity-matching model. Therefore, this technique allows for the correction of possible selection biases determined by a nonrandomized controlled trial design. In fact, when a good covariate balance between intervention and control groups was obtained, an estimate of the average treatment effect on the treated individual (ATT) was possible and accountable as a specific effect.

The demographic variables included in this study were age, sex, body height, and weight. Physiologic variables were major health risk factors for the general population, namely the body mass index (BMI), hypertension, diabetes mellitus, dyslipidemia, smoking habit, and chronic disorders such as asthma or allergies that may be triggered by environmental stimuli during nature experiences. Other physiologic variables considered as possible confounders of anxiety symptoms in the experimental setting were the physical exercise habit, any experience in meditation practices, and the session site type. The sociological variables considered were having children and the degree of satisfaction from the relationship with friends and relatives. Psychological covariates considered as possible confounders and modifiers were baseline long-term tendency to anxiety (STAI-T scores) and, due to the short-term nature of anxiety symptoms (measured with the STAI-S questionnaire as above described), other psychological covariates, such as baseline POMS depression, vigor, fatigue, confusion, anger, and self-esteem domain scores. Missing data in one or more of the aforementioned key covariates implied the exclusion of the participant from the final analysis.

### 2.5. Data Analysis

Microsoft Excel was used to organize the data collected. The software used for statistical analyses was “R” [65], using RStudio ver. 2022.07.2 [66], and the packages “matchit” [67], “marginaleffects” and “stats” [65]. The threshold for statistical significance of the overall effect size was set at *p* < 0.05.

Total MT air concentrations at any considered site were analyzed, and the average concentration value, equal to 0.20 µgm^−3^ (range 0.01 to 0.99 µgm^−3^), was established as a cut-off value between low and high MT exposure. 

Choosing the average concentration of the volatile substances analyzed allowed the partitioning of the whole sample of participants into two subgroups of comparable numerosity and, therefore, maximized the effectiveness of the propensity score matching. High MTs exposure was considered as the treatment of interest.

The Mann–Whitney nonparametric test was used to test whether differences in MTs and α-pinene concentrations measured in different landscapes were statistically significant [68].

One-way ANOVA tests were performed to compare the effect of different MT concentrations on the anxiety outcome and test for a statistically significant difference in anxiety levels between the exposed and unexposed groups.

Generalized linear model regression was performed to test for any correlation between the anxiety level outcome as the dependent variable and other continuous independent variables.

Then, the propensity score matching, which is a well-known statistical method especially used in medicine, epidemiology, and behavioral research [69,70,71,72,73], was performed to estimate the average marginal effect of the exposure to high MT air concentrations during forest therapy sessions on change-from-baseline STAI-S scores reported by the study participants and accounting for confounders and effect modifiers, as described above. A rule-of-thumb of at least 20 participants per included covariate was applied.

It was first decided to try with a 1:1 nearest neighbor propensity score matching without replacement and a propensity score estimated using logistic regression of the intervention on the covariates. This matching specification yielded an acceptable, yet not fully satisfactory, balance (standardized mean differences for the covariates <0.15), so it was instead decided to opt for an optimal full matching on the propensity score, which yielded a satisfactory balance. The propensity score was estimated using a logistic regression of the intervention on the covariates. Baseline covariates were considered balanced across the high and low MT exposure groups in the matched sample if the absolute difference of the standardized difference was ≤0.1.

The efficacy endpoint, which was a significant short-term reduction in STAI-S scores after a forest therapy session, was compared across the propensity-matched groups fitting a linear regression model with the change-from-baseline (post-test values minus pre-test values) STAI-S score as the outcome of interest and the intervention, covariates, and their interactions as predictors. This analysis also included the full matching weights in order to estimate the effect of the intervention and its confidence interval. A g-computation was then performed in the matched sample to estimate the average effect of the intervention on those who received it (ATT). A cluster-robust variance was used to estimate its confidence interval with matching stratum membership as the clustering variable.

An additional analysis was performed, using the propensity score matching to estimate the average effect of the exposure to above average α-pinene air concentrations (0.10 μgm^−3^) on change-from-baseline anxiety symptom scores reported by the study participants, accounting for confounding by the aforementioned covariates. The same procedure and matching methods, as described above, were applied with similar results, with the only difference being that the propensity score was estimated using a probit regression of the intervention on the covariates, which in this case yielded better balance than logistic regression (covariates standardized mean difference <0.1 vs. <0.15).

Other additional analyses were performed considering the 3rd quartile of total MT and α-pinene concentrations as the cutoff values for treatment exposure, in order to check for possible dose-dependent effects. Therefore, the propensity score matching was applied to estimate the average effect of the exposure to MTs (0.28 μgm^−3^) and α-pinene air concentrations (0.16 μgm^−3^) above the 3rd quartile on change-from-baseline anxiety symptom scores reported by the study participants, accounting for the aforementioned covariates. As with the previous analyses, for the propensity score of the above-the-3rd quartile MT concentration, the best achievable covariate balance was estimated using a logit regression of the intervention on the covariates, which included three covariates with a standardized mean difference of <0.12 instead of <0.1. This was considered acceptable, especially considering the high number of covariates involved. For the propensity score of the above-the-3rd quartile α-pinene concentration, the best achievable covariate balance was estimated using a probit regression of the intervention on the covariates, yielding a good balance with a standardized mean difference of <0.1 for all covariates. Finally, further similar analyses were conducted for above-average sabinene, o-cymene, and camphene concentrations exposures, yielding a good balance (standardized mean difference <0.1) for each test.

In the above-described cases, the efficacy endpoint was compared across the propensity-matched groups fitting a linear regression model with the change-from-baseline STAI-S score as the outcome of interest and the intervention, the above-mentioned covariates, and their interactions as predictors, including the full matching weights, in order to estimate the effect of the intervention. A g-computation was then performed in the matched sample to estimate the ATT of those who received the intervention, and a cluster-robust variance was used to estimate its confidence interval with matching stratum membership as the clustering variable.

## 3. Results

### 3.1. Preliminary Analyses

Overall, 655 participants were recruited, and 150 of them were excluded from the final analysis due to missing data in one or more key variables (i.e., demographic data and psychometric tests). The final sample consisted of 505 participants, with a higher share of females (65%). The distribution of participants in different age groups was as follows: 18–29 years (10%), 30–44 years (19%), 45–54 years (19%), 55–69 years (42%), and over 70 (10%).

Figure 3 shows the average and standard deviation of MT and α-pinene concentration levels in mountain, hill, pinewood, and urban park sites. A clear tendency toward lower concentration levels of both MTs and α-pinene in urban parks can be noted. However, due to the high variability and relatively low numerosity of hill, pinewood, and urban parks, no significant difference based on the Mann–Whitney test was found. Apparently, α-pinene accounted on average for nearly half of the total MT concentrations in each landscape.

A linear regression considering STAI-S scores observed after any forest therapy sessions as the dependent variable and total MT concentration as the independent variable, which resulted in an estimated effect of −3.24 STAI-S points (F = 4.497, *p* = 0.03), thus showing that on average any increase in MTs concentration was associated with a decrease in STAI-S scores. However, in consideration of the many possible confounding variables, such a result should not be interpreted as causation.

Table 1 shows the one-way ANOVA results for change-from-baseline STAI-S scores after exposure to high or low MTs and α-pinene air concentrations, with thresholds chosen as the average and the 3rd quartile concentrations, as well as the gender-based subgroup analysis. The tendency for all groups was towards anxiety symptom reduction after exposure to higher total MT and α-pinene air concentrations. The one-way ANOVA for different landscapes (mountain, hill, and urban parks) indicated a tendency towards anxiety symptoms reduction after exposure to higher total MT and α-pinene air concentrations. However, no such analyses could achieve statistical significance.

Since the one-way ANOVA could not account for confounding variables, the propensity-matching methodology was used to test for possible specific effects of MT exposure on anxiety levels.

Figure 4 shows the balance of covariates before and after propensity matching for exposure to high total monoterpenes air concentration. After matching, all standardized mean differences for the covariates were below 0.1, indicating adequate balance. Since full matching normally uses all units allocated to the intervention and control, no units were discarded.

Figure 5 shows the balance of covariates before and after propensity matching for exposure to high total α-pinene air concentration. After matching, all standardized mean differences for the covariates were below 0.1, indicating adequate balance. Since full matching was used as well, no units were discarded.

Figure 6 and Figure 7 show the balance of covariates before and after propensity matching for exposure to above-the-3rd quartile total MTs and α-pinene air concentrations, respectively [69,70,71,72,73].

### 3.2. Anxiolytic Effect of the Exposure to Monoterpenes and α-Pinene

Based on the propensity score matching analysis described in Section 3.1, the estimated effect on anxiety symptom levels, following the exposure to above average total concentration of MTs, was −1.28 STAI-S points (95% C.I. −2.51 to −0.06, *p* = 0.04), indicating that the average effect of the exposure to high MT air concentrations during forest therapy sessions was to decrease anxiety symptoms.

In the second analysis, focused on α-pinene, the estimated effect was −1.31 STAI-S points (95% C.I. −2.15 to −0.12, *p* = 0.03), indicating that the average effect of exposure to high α-pinene air concentrations was to decrease anxiety symptoms, confirming the previous result and suggesting a pivotal role of α-pinene in the anxiety-reducing effect.

In the third analysis, focused on above-the-3rd quartile MT concentrations, the estimated effect was −1.61 STAI-S points (95% C.I. −2.84 to −0.4, *p* = 0.01), indicating that the average effect of exposure to above-the-3rd quartile MTs concentrations in forests was to decrease anxiety symptoms.

In the fourth analysis, focused on above-the-3rd quartile α-pinene concentrations, the estimated effect was −1.68 STAI-S points (95% C.I. −3.2 to −0.16, *p* = 0.03), indicating that the average effect of exposure to above-the-3rd quartile α-pinene air concentrations was to decrease anxiety symptoms, suggesting that α-pinene may have a major role in the anxiolytic effects of inhaled MTs.

Finally, analyses conducted for investigating the effects of the exposure to above-the-average air concentrations of sabinene, o-cymene, and camphene on anxiety levels did not produce any statistically significant result, even after good covariate balance was obtained. However, it is interesting to note a trend toward statistical significance for above-average sabinene exposure (ATT = −1.226, *p* = 0.08).

Table 2 summarizes the above-reported results of the propensity score-matched analysis estimating the average effect on state anxiety (STAI-S) in the individuals exposed.

## 4. Discussion

This study showed for the first time a specific effect of the total concentration of inhaled MTs and, in particular, of α-pinene, on anxiety symptom levels of participants in forest therapy sessions at various sites. Although this was not a randomized controlled trial, the propensity matching methodology, applied to a quite large cohort of subjects involved in several identically structured forest therapy sessions in different sites, allowed an accurate balance for all participants’ variables, including their psychological baseline characteristics and involvement in experiences in urban parks or remote forests. This way, the participants could be partitioned into an intervention group, exposed to MT concentrations above a cut-off level derived from the available data, and a control group, exposed to lower concentrations.

The quite comparable effect sizes of STAI-S reduction associated with above-average total MTs and high α-pinene air concentrations accounted for around 28% of the average decrease in STAI-S observed across the entire sample (−4.6 points), which provided an estimate of the specific anxiolytic contribution of these volatile compounds. Moreover, the increased anxiolytic effect of MTs and α-pinene above the 3rd quartile of the respective air concentration may suggest a certain degree of dose dependency.

These results confirm the hypothesis formulated in the previous pilot study, where short-term anxiety improvements observed after forest exposure were likely associated with high levels of BVOCs in the air [14].

Inhalation of natural essential oils (EOs), of which MTs (including α-pinene) are important constituents, has long been the subject of clinical studies that showed significant effects on depression and anxiety disorders in humans, as well as of preclinical examinations, which confirmed significant effects on depression and anxiety-like symptoms in animal models [74,75].

On the one hand, the role of the cardiorespiratory system, responsible for the systemic absorption of volatiles and the subsequent delivery to organs, including the central nervous system, and, on the other hand, the direct stimulation of the brain through olfactory organs, were hypothesized as fundamental for explaining the therapeutic efficacy of inhaled EO molecules on mood disturbances. Olfactory stimuli can directly affect mood, through direct anatomical and functional links of the olfactory system with the limbic system. However, more precise mechanisms of action can be revealed only by experiments with isolated EO constituents, including individual MTs such as α-pinene [74].

The results of this study allow the identification of a ubiquitous class of components of the atmosphere in green areas, namely MTs, as specific therapeutic agents with regard to the levels of anxiety symptoms. Moreover, our results show that α-pinene may play a major role in the anxiolytic effects of inhaling MT-rich air, leaving open questions about possible specific effects of other monoterpenes. These findings substantiate the emerging concept of the environment as a “healing environment”, a source of valuable elements actively promoting human health and well being, and not only as a resource to be protected and preserved from pollution [76].

The elucidation and quantification of a specific healing mechanism of forest environments and green areas can offer a powerful tool to direct the widespread efforts to plan and optimize forest therapy trails, often relying on the prediction of MT concentrations based on plant species and meteorological variables [31,32,34,77].

Studies documenting the effects of forest immersion experiences on anxiety symptoms and stress, such as those analyzed in recent comprehensive reviews [78,79], could be partially reinterpreted in light of our study findings, and future experiments should take these results into adequate account. As well, further studies might consider investigating the effects on anxiety symptoms of MT or α-pinene inhalation duration, the respective efficacy thresholds, and possibly the effects of MTs (also other than α-pinene) on physiological aspects.

Finally, this growing body of evidence indicates that green environments have an active and specific effect in reducing the burden of anxiety symptoms in the general population, thus impacting not only mental but also, indirectly, cardiovascular health, as shown in recent large cohort studies [80], which implies great potential in terms of saving public health expenditure. Therefore, advocacy for the encouragement of protecting and creating green areas should be addressed.

The main limitations of this study were the lack of a randomized controlled trial design and a significant drop-out rate from the final analysis due to suboptimal control over the participants’ compliance with filling out the self-report questionnaires. Another minor limitation consisted of the gender imbalance, with the dominance of female participants over males. These limitations should be considered for future studies on the topic.

## 5. Conclusions

The results of this study showed for the first time that inhalation of plant-emitted MTs, and in particular α-pinene, can produce a specific anxiolytic effect. On the basis of this evidence, as a practical recommendation to help manage anxiety symptoms, it can be suggested to regularly visit sites whose air is rich in MTs such as α-pinene. In this regard, in order to identify proper sites, it would be important to measure these BVOCs in the air, or at least to identify areas where there are tree species that are likely to release large quantities of these substances in the atmosphere. Inhaling these natural compounds during a 3-h outdoor relaxing session can improve anxiety levels and positively contribute to the individual’s well-being.

Our findings highlight the importance of focusing on the environment as a source of health-promoting factors, with the necessity to promote appropriate policies and raise awareness about environmental issues. Further perspectives are to investigate the possible effects of inhalable forest MTs on physiological outcome measures.

## Figures and Tables

**Figure 1 ijerph-20-02773-f001:**
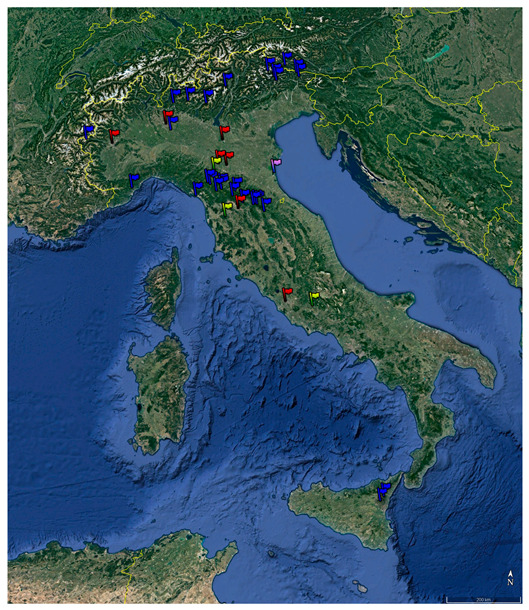
Location of the experimental sites. Blue: remote mountain; yellow: hill; violet: remote coastal pinewood; red: urban park.

**Figure 2 ijerph-20-02773-f002:**
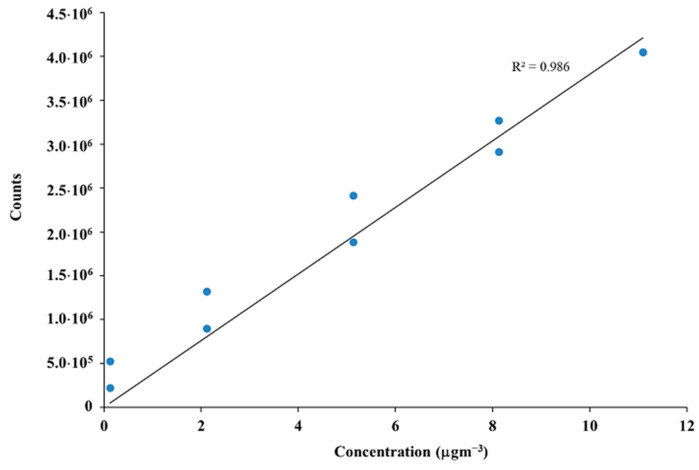
Calibration curve for α-pinene.

**Figure 3 ijerph-20-02773-f003:**
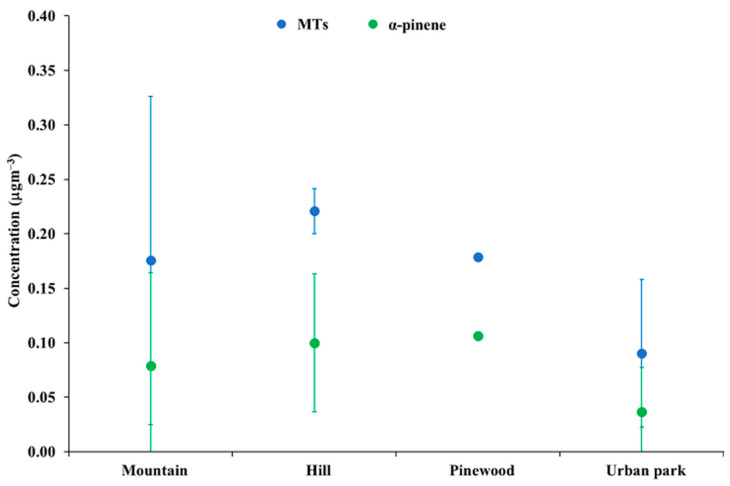
Average and standard deviation of MT and α-pinene concentration levels in mountain, hill, pinewood, and urban park sites.

**Figure 4 ijerph-20-02773-f004:**
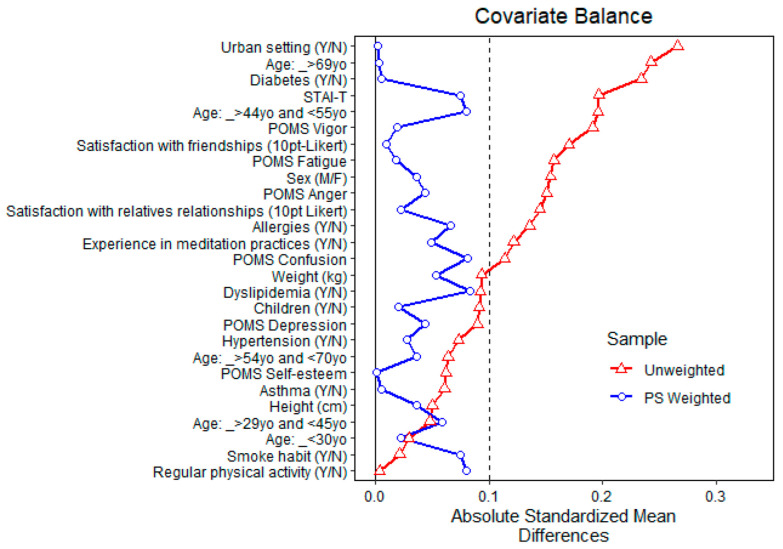
Balance of covariates before and after propensity matching (PS weighted) for exposure to high (above the average) total monoterpenes air concentration.

**Figure 5 ijerph-20-02773-f005:**
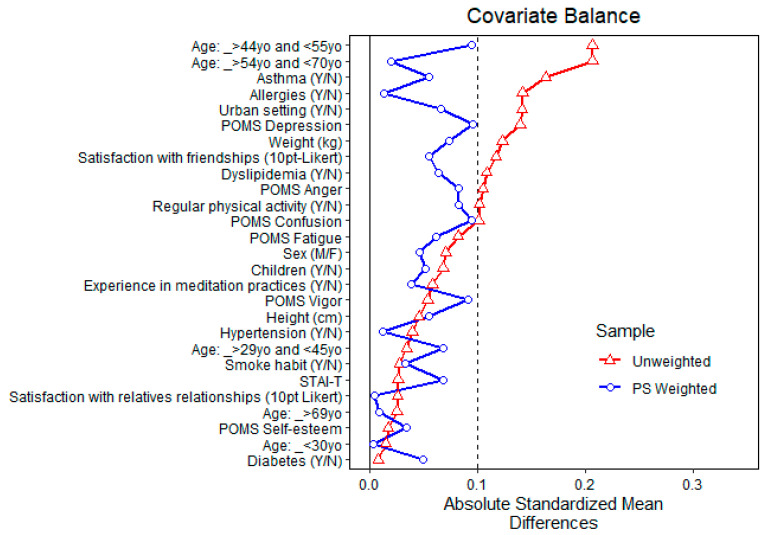
Balance of covariates before and after propensity matching (PS weighted) for exposure to high (above the average) α-pinene air concentration.

**Figure 6 ijerph-20-02773-f006:**
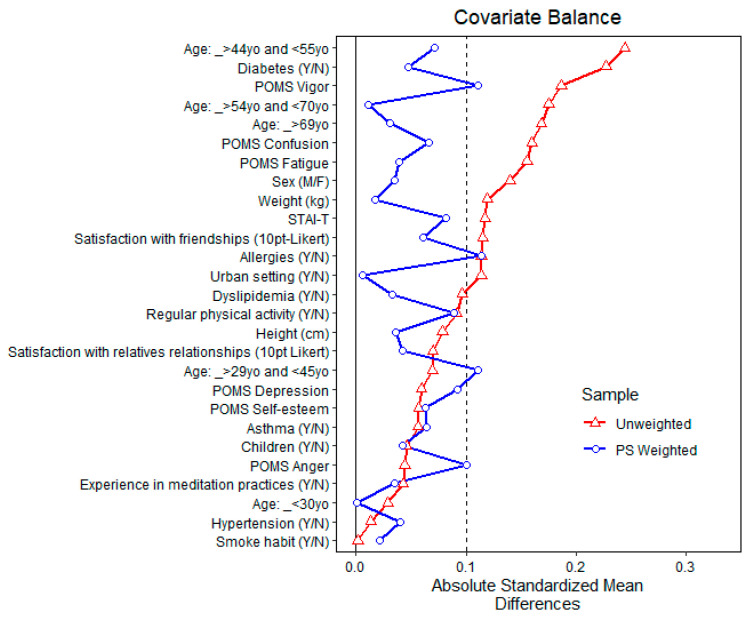
Balance of covariates before and after propensity matching (PS weighted) for exposure to above-the-3rd quartile total monoterpenes air concentration.

**Figure 7 ijerph-20-02773-f007:**
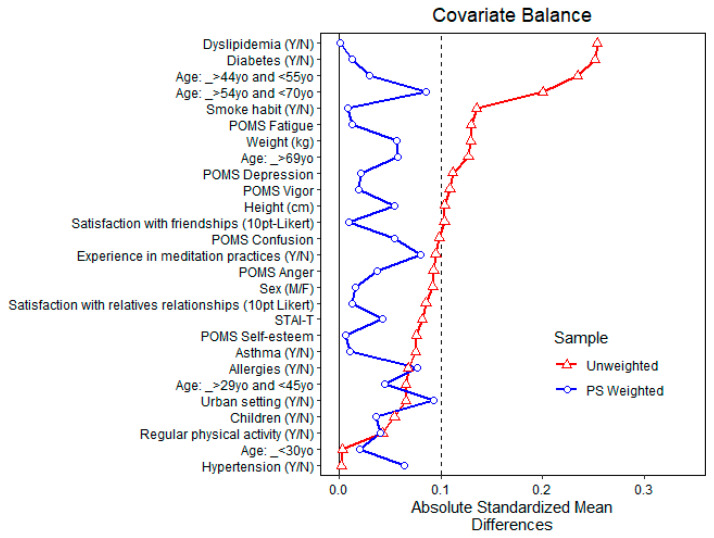
Balance of covariates before and after propensity matching (PS weighted) for exposure to above-the-3rd quartile α-pinene air concentration.

**Table 1 ijerph-20-02773-t001:** Results of the one-way ANOVA for exposure of the whole group and the subgroups of females and males to MT and α-pinene air concentrations above the average and above the 3rd quartile on change-from-baseline STAI-S scores. Significant results are indicated by *.

Quantity	Threshold ^1^	Population	F Statistics	*p* Value
MTs	Average	Total sample	3.93	0.05
MTs	3rd quartile	Total sample	4.50	0.03 *
α-pinene	Average	Total sample	3.67	0.05
α-pinene	3rd quartile	Total sample	4.26	0.04 *
MTs	Average	Females	3.71	0.05
MTs	Average	Males	3.24	0.07

^1^ Threshold for high exposure.

**Table 2 ijerph-20-02773-t002:** Results of the propensity score-matched analysis estimating the average effect on state anxiety (STAI-S) in the individuals exposed (ATT). Significant results are indicated by *.

Condition	Best Covariate Balance	Matched (n)	Unmatched (n)	ATT	95% CI	*p* Value
Exposed to an above-the-average MT air concentration	Std. Mean Difference < 0.1	Controls = 268 Cases = 237	Controls = 0 Cases = 0	−1.28	−2.51 to −0.06	0.04 *
Exposed to an above-the-average α-pinene air concentration	Std. Mean Difference < 0.1	Controls = 296 Cases = 209	Controls = 0 Cases = 0	−1.31	−2.15 to −0.12	0.03 *
Exposed to an above-the-3rd quartile MT air concentration	Std. Mean Difference < 0.12	Controls = 356 Cases = 149	Controls = 0 Cases = 0	−1.61	−2.84 to −0.4	0.01 *
Exposed to an above-the-3rd quartile α-pinene air concentration	Std. Mean Difference < 0.1	Controls = 345 Cases = 160	Controls = 0 Cases = 0	−1.68	−3.2 to −0.16	0.03 *

## Data Availability

All data are available by contacting the corresponding author.

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
