# Peer review of "Effects of Plant-Emitted Monoterpenes on Anxiety Symptoms: A Propensity-Matched Observational Cohort Study"

_ijerph, 2023, doi:10.3390/ijerph20042773_

Round 1

Reviewer 1 Report

Thank you for the opportunity to review the manuscript „Effects of Plant-Emitted Monoterpenes on Anxiety: A Propensity-Matched Observational Cohort Study“. The aim of this study was to investigate whether inhaling plant-emitted biogenic volatile compounds, namely monoterpenes (MTs), can produce specific effects on anxiety.

Personally, I was very interested in reading this manuscript.

Although this study is relevant, I have some concerns:

Major concerns

Abstract: Indicate the study’s design with a commonly used term in the abstract.

Materials and Methods: Clearly define all outcomes, exposures, predictors, potential confounders, and effect modifiers. Describe any efforts to address potential sources of bias. Explain how quantitative variables were handled in the analyses. If applicable, describe which groupings were chosen and why.

Statistical methods: Describe all statistical methods, including those used to control for confounding. Describe any methods used to examine subgroups and interactions.

Results: This is the most confused part of this manuscript. Insufficient data analysis in the results section in the current format (for example, the results analysed must be presented in tabular form). I propose that the results obtained as well as the statistical indicators might be presented in the tables.

Discussion: Lines 281-285: This is an override of results.

Additionally, summarise key results with reference to study objectives, please. Give a cautious overall interpretation of results considering objectives, limitations, multiplicity of analyses, results from similar studies, and other relevant evidence. Discuss the generalisability (external validity) of the study results.

Conclusions: The manuscript seems to lack a precise description of the practical recommendations.

Minor concerns

The term “anxiety” should be replaced throughout the paper by “anxiety symptoms”. Unless the authors have established and confirmed clinical diagnoses of study participants.

The results section must be described in the past tense.

Line 350: I propose that the date of authorisation must be indicated.

Best Regards

Reviewer 2 Report

The MS considers positive influence of monoterpenes (in particular, α-pinene) which belong to the vast family of biogenic volatile organic compounds (BVOCs) on reducing human anxiety. No doubts, that topic is very actual and has an apparent practical importance. The research is based on results of comprehensive experimental work that included both BVOC concentration measurements and examination of psychophysical state of more than 500 participants at several sites. The analysis of data resulted in conclusion (that is quite predictable) about positive influence of monoterpenes on anxiety. The paper contains informative introduction and detailed description of experimental methods what can’t be said about results presentation.

To my opinion, results of such complicated experiment that of course is a strong part of research should be presented and analyzed more fully. In fact, only one figure was included to illustrate the results of experiment that is not sufficient.

First, the experiment was carried out in different landscapes as shown on Fig. 1. It would be interesting to check if it implies the results. For that it’s worth to split obtained data into groups (for instance, mountain forest sites, plain forest sites and urban parks).

Second, both initial experimental data on MT measurements and cohort survey must be reflected in the MS somehow (at least, as a supplementary materials). It can be the plots (or tables) of MT and α-pinene concentrations with minimal statistics, or diagrams with values averaged by landscapes. This is important both for scientific analyses and initial data verification.

Third, correlation between MT level with anxiety characteristics (STAI-S score) should be shown.

Fourth, despite the gender disproportion in the observational cohort it is worth to consider results depending on gender.

 Below are also few remarks attributed to the specific places in the MS.

 Remarks

Lines 193-194 – calibration curves ought to be presented at least in supplementary materials. Confirmation of the measured data reliability is very important.

Lines 208-210 - How is justified the choice of the threshold MT value? Is it really indifferent for the experiment whether the concentration of MT is 0.2 μg/m-3 or five times higher? It would be interesting to study whether MT highest level implies the anxiety more than average ones.

Figure 2 – what means the dash vertical line?

 Lines 267-270 – why the results about α-pinene are not illustrated? At least correlation with MT and plot like one on the Fig. 2 would be interesting. The expected outcome maybe that it is enough to measure α-pinene as the main MT agent to assess MT’s medical effect.

Round 2

Reviewer 1 Report

Dear Authors,   Welcome to a well-prepared Article. Thank you for changing your manuscript. I recommend the paper for acceptance.   Kind Regards

Reviewer 2 Report

I guess that authors performed substantial work to modify the manuscript. To my opinion, it has been significantly improved and can be published in current view